# Identification of métiers in a multi-gear, multi-species fishery

**Pedro Leitão**[1,2]*, **Margarida Castro**[1], **Aida Campos**[1,2]

1 Centro de Ciências do Mar do Algarve (CCMAR/CIMAR LA), Campus de Gambelas, Universidade do Algarve, Faro, Portugal, 2 IPMA, Instituto Português do Mar e da Atmosfera, Rua Alfredo Magalhães Ramalho 6, Algés, Portugal

* pedro.leitao@ipma.pt

## Abstract

Accurate, gear-specific data is essential for fisheries management to quantify fishing pressure and ecosystem impacts, particularly in multi-gear fleets. In this study, we used electronic logbooks and sales notes from 2014 to 2023 to analyze the Portuguese multi-gear coastal fleet, obtaining information at a haul level. The objective was to identify and validate métiers down to level 6 according to the European Commission's definition (a combination of gear specifications, target species, fishing area, and season). This fleet comprises approximately 500 vessels using various types of traps, nets, and longlines to catch around 200 species of fish and invertebrates. We identified 28 métiers across six main gear types. The most representative métier is the octopus (*Octopus vulgaris*) trap fishery, followed by the European hake (*Merluccius merluccius*) gillnet and the black scabbardfish (*Aphanopus carbo*) longline fisheries. A strong association was found between most métiers and variables such as fishing season, fishing area, depth, and sediment type. Traps and nets were mostly used in the northwest area, in mixed sediments or sand, whereas longlines were mostly operated in the central west area, in steeper slopes on sandy and rocky bottoms. The study highlights the complexity of quantifying gear-specific fishing effort in multi-gear fisheries, where technical interactions occur, and competitive relationships exist between different fishing fleets and gears that exploit the same fish stocks.

## 1. Introduction

Fisheries dependent data – such as the type, amount, and size of fishing gear used, as well as associated catches – are essential for estimating fishing pressure. This metric is critical for assessing the environmental impacts of fisheries and informing management decisions [1,2]. In many countries, including those in the European Union, fishers are required to fill logbooks, which can be handwritten or electronic, to record information on fishing operations as part of the fisheries control policy [3]. This

**Data availability statement:** The data used for this study including electronic logbooks and sales notes can not be shared due to a non-disclosure agreement but is available on request to the Directorate-General for Natural Resources, Safety and Maritime Services at https://www.dgrm.pt/pnrd or by contacting Suzana Cano (sfcano@dgrm.pt). The data set is described in full within the paper and the R script used is shared as a Supporting Information file.

**Funding:** This study received Portuguese national funds from FCT - Foundation for Science and Technology through projects UIDB/04326/2020 (https://doi.org/10.54499/UIDB/04326/2020), UIDP/04326/2020 (https://doi.org/10.54499/UIDP/04326/2020) and LA/P/0101/2020 (https://doi.org/10.54499/LA/P/0101/2020), P. Leitão was the recipient of a PhD scholarship 2022.11214.BDANA. The funders had no role in study design, data collection and analysis, decision to publish, or preparation of the manuscript.

**Competing interests:** The authors have declared that no competing interests exist.

information helps estimate fishing effort and, depending on the availability of georeferenced data, determine its spatial distribution [4,5].

In multi-gear fishing fleets, mostly operating fixed gear (traps, nets, and longlines), quantifying gear-specific fishing effort and ecosystem impacts is challenging because most vessels use several types of gear with a wide range of gear characteristics such as net mesh size, hook size, and gear dimensions, making data collection for the multi-gear fleet considerably more complex than for mobile gears such as trawls or purse seines [6–8]. Moreover, the multi-gear fleet catches hundreds of target and non-target species. Certain gear types, such as trammel nets and gillnets, indiscriminately catch a wide diversity of species and size ranges [7–9], while others, such as longlines, often target a limited group of species and size ranges, being therefore more selective [7,8,10]. Furthermore, vessels often use multiple gear types on a single trip, and the same species may be targeted by different gears [11–13], adding complexity to data collection programs.

At the European level, characterization of multi-gear fleets is particularly important for southern countries, where the number of active vessels using fixed gears is much higher than in northern countries. Based on the Fleet Register data [14], these numbers are approximately 2,700; 7,000; 3,900; 1,800; and 10,000 for Portugal, Spain, France, Italy, and Greece respectively, compared to the United Kingdom, Ireland, Belgium, Denmark, and Netherlands, where such numbers are lower (respectively 350; 1,400; <100; 1,400; and 300).

The Portuguese multi-gear fleet, including the local and coastal fleets, comprises over 90% of the total number of vessels in the country, accounts for 36% of the total catch of commercially exploited species, and 64% of the total value landed [15], which makes it very important socially and economically. In 2024, the main species landed by this fleet (in weight) were tuna (*Thunnus spp*), octopus (*Octopus vulgaris*), black scabbardfish (*Aphanopus carbo*), Atlantic chub mackerel (*Scomber colias*), and cuttlefish (*Sepia officinalis*) [15]. In the coastal segment of the Portuguese multi-gear fleet, which comprises about 500 vessels between 9 and 35 meters in length, about one third are equipped with electronic logbooks [16], recording information at the haul level, including the catch (main species and corresponding weights), gear type, location, and timestamp of setting and hauling. This fleet employs a diverse range of fishing gears, which differ depending on the target species, the geographical area, the time of year and the landing port [17]. For instance, black scabbardfish and monkfish are primarily caught in central and southern Portugal using bottom longlines and gillnets, respectively, while cuttlefish trammel net landings are significantly higher in the early months of the year [17].

Previous gear selectivity studies in trammel net fisheries have identified different fisheries and quantified their respective bycatches [9]. However, the lack of a systematic program for collecting data on all important fisheries over time, limits our understanding of the true scale of the multi-gear fleet's impact on the ecosystem. With the advent of big data access and storage, algorithms have emerged as valuable tools for identifying fishing patterns and predicting the gear used in multi-gear fisheries [12,18]. Researchers analyzed fishing trip records from the Portuguese coastal

multi-gear fleet, provided in electronic logbooks and sales notes, along with fleet characteristics [12]. The main fishing gears used were predicted using decision trees with an overall classification error of 14% [12]. However, neither the target species nor the technical details of the gear were available for their study, preventing the identification of métiers down to the level 6 [19], and consequently their respective fishing effort and environmental impacts.

According to the European Commission Data Collection Framework, a métier is defined as "a group of fishing operations targeting a similar (assemblage of) species, using similar gear, during the same period of the year and/or in the same area, and characterized by a similar exploitation pattern" [19]. The identification of métiers in the multi-gear fleet improves existing knowledge about fishing dynamics, allowing the quantification of the fishing effort as well as its temporal and spatial distribution [7]. It also supports the development of management and conservation measures that may include environmental protection, such as the spatial and temporal closure of areas for specific gears, without substantially affecting the whole fishing activity.

Several studies have addressed methodological options and related issues with a view to identifying target species and métiers in the context of multi-gear, multi-species fisheries. A commonly used method is clustering analysis, such as hierarchical k-means clustering, and unsupervised machine learning analysis, which do not require prior knowledge of the classification variable used to define the groups. This approach has been applied to sales notes, fishing licenses, and on-board sampling to identify target species or fleet segments [16,20–22]. An intrinsic limitation of this clustering technique is the requirement to predefine the number of groups to a value equal to or less than 30, in the absence of information on the potential number of métiers [20,23–25], forcing the user to choose a cut-off value based on either high intra-cluster similarity, resulting in many irrelevant métiers, or low intra-cluster similarity, resulting in missing relevant métiers [22]. On a smaller scale, one study used similarity percentages species contribution analysis (SIMPER), classified trip types, and applied the results to sales notes to identify métiers and characterize fisheries [7]. Due to its high computational requirements, this method is rarely used in big data scenarios [7].

The analysis of interviews, combined with fishing licenses and sales registers from fishing trips, has been used to identify métiers in a small-scale multispecies fishery [26]. In this simple and direct approach, the use of métiers over time allowed the identification of new, as well as past métiers, at the regional level. Researchers used a similar approach and validated a total of 10 métiers of the multi-gear fleet in southern Portugal [13]. However, this approach would be impractical as a long-term, nationwide strategy for obtaining information on gear-specific fishing effort in a multi-gear scenario. This is because it would require intensive sampling, given the differences in seasonality and the large number of ports to be covered.

Previously, the comprehensive analysis of logbook data was only used to identify the main gear [12]. In the present study, sales notes and electronic logbooks from the period 2014–2023 were used to identify métiers of the multi-gear coastal fleet down to Level 6 [19]. The approach combines the type of gear and its characteristics (such as mesh or hook size) with the target species at the haul level. The target species were assumed to be those that attain a higher sale value. Subsequently, the resulting métiers were validated through previous studies, onboard observations and interviews with fishers. Lastly, a selection of the most important validated métiers was analyzed for seasonal and spatial trends.

## 2. Material and methods

### 2.1 Study area

This study focuses on the multi-gear coastal fleet operating in the waters of the Portuguese continental shelf, in southwestern Europe. This area is characterized by the existence of several canyons and strong currents, resulting in a diversity of ecosystems and bottom types [27]. Based on previous studies describing canyons, currents, and prevailing winds [20,28,29], the coastline was divided into four geographical areas: 1) northwest, 2) center west, 3) southwest, and 4) south (Fig 1). The northwest and center west areas are separated by the deep Nazaré Canyon, while the center west and

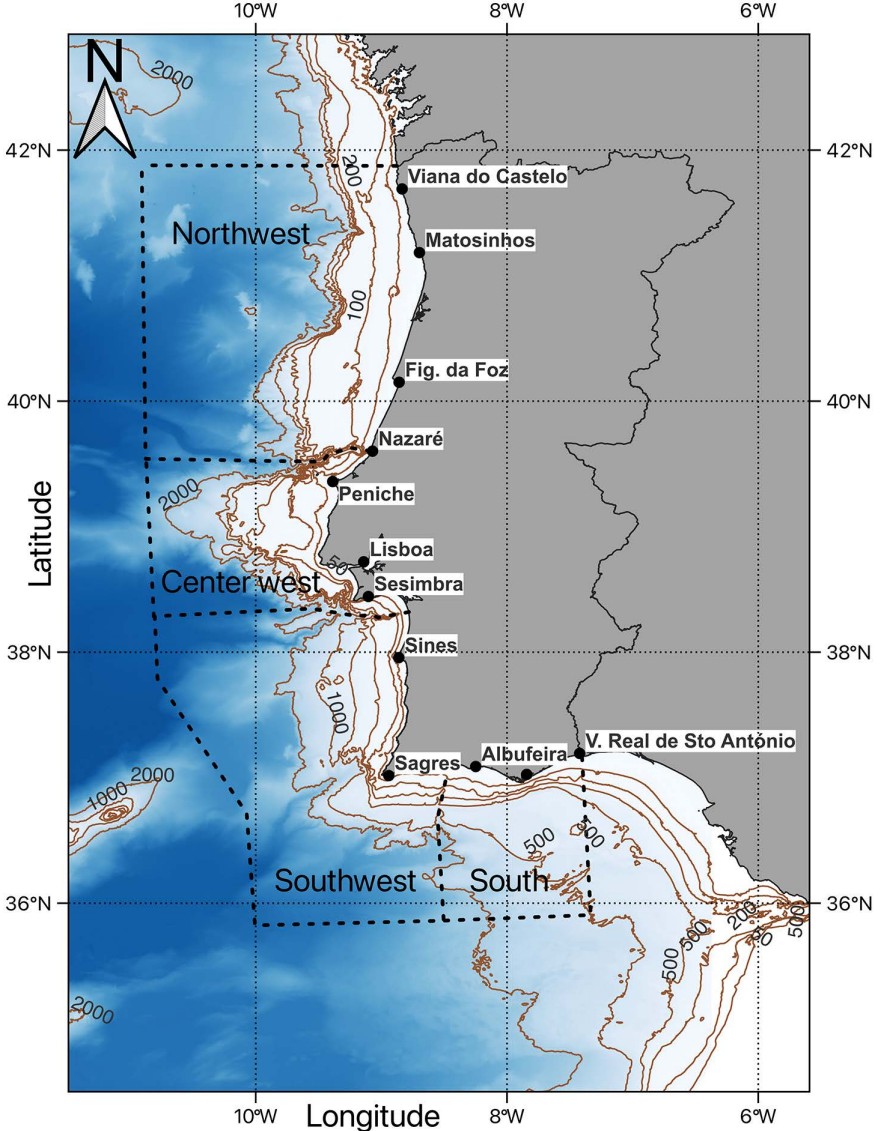

**Fig 1. Map of the Portuguese continental coast.** It shows the main landing ports and landmarks, and four distinctive fishing areas: northwest, center west, southwest, and south. Isobaths at 50, 100, 200, 500, 1,000, and 2,000 meters are visible.

southwest areas are divided by the Setubal Canyon, south of Sesimbra. These different areas are home to more than 200 commercially exploited species [30].

## 2.2 Multi-gear coastal fleet

The Portuguese multi-gear coastal fleet operates in a variety of environments, ranging from the continental shelf to the open sea [16]. The fleet consists of around 500 vessels measuring over nine meters in length and uses nets, traps, and longlines to catch a wide variety of commercially exploited species. Fishing trips can last from hours to weeks, depending on the vessel's fishing strategy [12]. Most vessels over 12 meters in length fill electronic logbooks that provide information of the fishing gear used, location of each haul and amounts captured for the different species. This haul-level information

can be linked to trip sale notes to determine the value of species caught [12] and used to identify potential métiers and to assess seasonal and geographical patterns of fishing effort.

## 2.3 Databases used

The data used in the study consisted of electronic logbook records and sales notes from 2014 to 2023, available for approximately 150 vessels, about one-third of the multi-gear coastal fleet. When filled out at a haul level, electronic logbooks provide information on individual vessel operations, including location (geographical coordinates), timestamps for setting and hauling, main gear used (bivalve towed dredges, traps, gillnets, trammel nets, drifting longlines, and bottom longlines), and main species caught in each haul (by weight). Sales notes (at the trip level) include date, landing port, and species sold (weight and average price). Average prices from the sales notes enabled the estimation of the contribution of each species to the value of the haul.

Additional information was associated with each haul based on its location (midpoint of setting and hauling), including depth [31] and sediment type [32]. This information was rasterized using a high level of detail (with a pixel size of less than 0.01 units or 100 meters). Three depth strata were considered: shallow, less than 200 meters; medium, 200–500 meters; and deep, over 500 meters. Five sediment types were used based on the European Marine Observation and Data Network datasets: mud, sand, coarse grain sediment, mixed sediment, and rock. Hauls with an unknown type of sediment were classified as "Unknown". The depth and sediment type for each haul were assigned using the Sample Raster Values tool, and the resulting file was exported to RStudio.

## 2.4 Identification of potential métiers

In previous studies, the species with the highest value contribution to the catch was often considered the target species [7,26,33]. Value, not weight, is important for the identification of métiers that target less abundant catches of very valuable species, such as lobsters. Following this approach, the métier associated with each haul was defined as a combination of the gear (including gear details such as mesh and hook size) and the most valuable species caught, which was assumed to be the target species.

The selection of hauls for the multi-gear coastal fleet, used for the identification of métiers, involved the following steps:

1. Selection of hauls in Portuguese waters (FAO subarea 27.9).

2. Exclusion of hauls associated with depths greater than 2,000 meters (outside the area of operation for the fleet of interest); exclusion of hauls associated with trips that did not depart from and return to a port on the Portuguese mainland.

3. Exclusion of hauls with errors or incomplete information, such as reference to miscellaneous fishing gear, zero catches, landing price unavailable, and abnormally high reported weights (i.e., reported weight of hauls sometimes exceeded the maximum sold weight for that species).

The hauls removed in steps 1–3 corresponded to 1.8% of the total number, leaving 197,081 valid hauls. A total of 307 potential métiers were identified (S1 Table). The data summary revealed varying activity levels across the fishing gears analyzed, with the highest in nets and traps, and the lowest in longlines and dredges (Table 1).

A strategy was implemented to reduce the number of potential métiers to work with, since considering 307 was unrealistic. For each main gear, a bar graph was created, ranking métiers from most to least frequent. Based on visual analysis, the point at which the graph curvature stabilized was used to decide which métiers were more important and would be considered for validation. The applied criteria aimed to select the most relevant métiers while maintaining the proportional importance of the chosen ones to the main gear to which they belong. In this way, even less represented gears, such as drifting longlines, had métiers that were selected.

**Table 1. Main gears and respective number of potential métiers, along with vessels, trips and hauls involved. * Total numbers of potential métiers and hauls correspond to the sum of columns, while the remaining correspond to unique vessels and trips. This happens since vessels use multiple gears, and trips may have multiple hauls.**

| Main gear | | Number of potential métiers | Number of vessels | Number of trips | Number of hauls |
|---|---|---|---|---|---|
| Gear FAO code | Name | | | | |
| DRB | Bivalve dredges | 7 | 3 | 1 062 | 2153 |
| FPO | Traps | 63 | 103 | 44 799 | 57 261 |
| GNS | Gillnets | 81 | 107 | 28 576 | 38 508 |
| GTR | Trammel nets | 93 | 111 | 52 563 | 75 888 |
| LLD | Drifting longline | 12 | 28 | 556 | 879 |
| LLS | Bottom longline | 51 | 64 | 18 094 | 22 392 |
| Total* | | 307 | 170 | 110 743 | 197 081 |

## 2.5 Validation of métiers

The most important métiers selected were validated using different approaches. First, the métiers that were well described in the literature were identified. A second criterion was validation through on-board observations carried out by trained observers in the previous couple of years, within the scope of the project Tecpescas. For the remaining métiers, specific interviews were conducted with fishers in ports of interest or over the telephone. Respondents were asked: "To the best of your knowledge, is species X targeted using fishing gear Y?" if the answer was yes, the métier was validated. Further characterization of the validated métiers was achieved by adding gear-selective characteristics (e.g., mesh size for nets and traps, or hook number for longlines), as well as associating them with a particular fishing area and/or season. Gear-selective characteristics were sourced from the Tecpescas project, while location and landing dates were obtained from electronic logbooks.

## 2.6 Data analysis

The number of hauls was considered a proxy for fishing effort, and its distribution across the different levels of the variables of interest (year, season, fishing area, depth, and sediment type) was investigated for each métier. A descriptive approach was used for all these variables, consisting of estimating the proportion of hauls at each level of the variable of interest.

Within each validated métier, chi-square goodness-of-fit tests were used for the variables: geographic location (fishing area), season, and depth (null hypothesis: uniform fishing effort distribution across all levels of the variable). Bonferroni correction for multiple tests was applied. The test was not applied in situations where the frequency of one or more levels was less than five (usually zero), as this is a requirement for the correct use of the test. For the discussion, in these situations, the distribution of fishing effort was considered non-uniform. When dealing with large samples, the absence of a métier in a season or area, for example, clearly corresponds to an intentional decision of not fishing, equivalent to rejecting the null hypothesis of uniform distribution of effort. Sediment type was not considered because there were many levels (five types) and hauls with unknown sediment. A chi-square test of independence was performed for the two most important variables in describing the métier: fishing area and season. As before, the test was not applied to data with frequencies less than five, and the association between these variables (whether positive or negative) was considered when discussing the results. Linear regression analysis of the total number of hauls per year for each métier was used to identify potential trends in their use. Rejection of the null hypothesis was considered with p-values less than 0.01 for all statistical tests.

Finally, heat maps were used to visualize the distribution of fishing areas for métiers targeting the same species, in order to determine if they shared the same fishing grounds.

Data analysis was carried out with the help of RStudio version 2023.09.1+494 and R version 4.4.1 [34] with packages "dplyr" [35], "plyr" [36] "reshape2" [37] "ggplot2" [38] "purrr" [39], "stringr" [40], "MASS" [41] and QGIS version 3.34.9-Prizren (S1 File).

## 3. Results

Out of the 307 potential métiers, the 37 most important ones, based on their significance in terms of haul count, were selected (Table 2, S1 Fig). The 270 potential métiers that were excluded had minimal representation, with only 6% of the total number of hauls, averaging 4.5 hauls per métier per year (0.1 to 86.5 hauls per year). Of the 37 métiers selected for analysis, a total of 28 were validated and characterized (Table 2). Sixteen métiers were validated because they were well described in published scientific papers, which are referenced in the validation column of Table 2. Three were identified based on observations made by the authors of this study while on board. The remaining métiers required specific interviews with captains of vessels involved in the fishery, conducted either over the phone or in ports. Nine métiers were identified this way (the number of interviews for each métier is mentioned in the validation column of Table 2). Nine other métiers could not be validated.

The results of the applied statistical analysis are included in S3 Table. The p-values from the goodness-of-fit test according to fishing area, season, and depth, showed that the null hypothesis of uniform distribution of hauls was rejected for most variables and métiers. This indicates clear differences in fishing intensity concerning these variables. All métiers showed differences with respect to fishing area and depth. In two instances, the null hypothesis with respect to season failed to be rejected: bivalve dredges and black scabbardfish bottom longlines (p-values 0.523 and 0.012, respectively). This indicates that these fisheries are practiced year-round. With respect to the independence of fishing area and season, the test was not rejected for pouting traps, black scabbardfish, and wreckfish bottom longlines (p-values 0.075, 0.026, and 0.097, respectively). This indicates that the seasons have the same importance in all areas and, inversely, the use of the areas is similar in all seasons.

The métier with the highest number of hauls of the multi-gear coastal fleet under study is the octopus (*Octopus vulgaris*) trap fishery, which is operated year-round, particularly along the northwest coast (Table 2). This is followed by the European hake (*Merluccius merluccius*) gillnet fishery in spring and summer, focusing on the northwest coast, and the black scabbard (*Aphanopus carbo*) fishery, which operates year-round, mainly on the central west coast. Overall, the trammel net fisheries, targeting soles (Soleidae), John Dories (Zeidae), monkfish (Lophiidae), and hake, among others, are those with stronger seasonal variability. All these fisheries, except the black scabbard fishery, operate mainly in shallow waters (less than 200 meters) on the continental shelf, in areas with sandy, muddy, or mixed sediments. Longline fisheries, which mostly operate in the central west area at medium or high depths, represent a smaller share.

Most métiers showed increased activity at certain times of the year, which was particularly evident in some cases, with over 50% of the hauls occurring in a single season. Examples of the latter include the common cuttlefish (*Sepia officinalis*) and European seabass (*Dicentrarchus labrax*) trammel net fisheries, which operate in winter, and the swordfish (*Xiphias gladius*) and blue shark (*Prionace glauca*) drifting longline fisheries, which operate in autumn and summer, respectively (S2 Table). Significantly increasing trends over the study period were observed for Norway lobster (*Nephrops norvegicus*) traps, turbot (*Scophthalmus spp*) with trammel nets, and swordfish (*Xiphias gladius*) drifting longlines (S3 Table).

Métiers are strongly linked to specific depths, as over two-thirds of the hauls (64%) occur at a single depth range (S2 Table). Most traps, gillnets, and trammel nets fisheries (75%) operate almost exclusively in shallow waters (less than 200 meters depth). Red porgy (*Pagrus pagrus*) and European conger (*Conger conger*) bottom longlines are also used in shallow waters. Silver scabbardfish (*Lepidopus caudatus*), blackspot seabream (*Pagellus bogaraveo*), European hake, and alfonsinos (*Beryx spp*) are caught with bottom longlines at depths of up to 500 meters. Swordfish drifting longlines, and black scabbardfish bottom longlines in deep waters.

**Table 2. Characterization of the 37 potential métiers sorted by descending order of their number of hauls. Unvalidated métiers highlighted in grey. Season – Winter: Jan to Mar, Spring: Apr to Jun, Summer: Jul to Sep, Autumn: Oct to Dec. Fishing Area – CW: center west, NW: northwest, S: south, SW: southwest. Depth – Shallow (less than 200 meters), Medium (200 to 500 meters), Deep (more than 500 meters). Sediment – Mud: Mud and sandy mud (very fine particles), Sand: Sand and muddy sand (fine, gritty particles), Mixed: Mixed sediments (combination of particle sizes); Rock: Rock, boulders or cobbles (hard substrate) or Unk: Unknown. \*The common octopus traps fishery includes non-baited shelter traps. Dashes (-) represent the variables of the unvalidated métiers that were not analyzed. Summary table based on appendix S2.**

| Gear | Name of métier | Number of hauls | Mesh size/hook number | Season | Fishing area | Depth | Sediment | Validation through |
|------|----------------|-----------------|------------------------|--------|--------------|-------|----------|--------------------|
| DRB | Solid surf clam bivalve dredges | 1 759 | 30 | All year | NW | Shallow | Sand | [42] |
| FPO | Common octopus traps | 48 390 | 8–50 or none* | Autumn to Winter | NW | Shallow | Sand and Mixed | [13] |
| FPO | Pouting traps | 2 697 | 50 | Spring to Summer | NW | Shallow | Sand and Mixed | Interviews (4) |
| FPO | Norway lobster traps | 2 461 | 30–50 | Summer to Autumn | CW; SW | Medium to Deep | Mud and Sand | [43] |
| FPO | European conger traps | 2 038 | 50 | Spring to Summer | NW | Shallow | Mixed and Rock | Interviews (3) |
| GNS | European hake gillnets | 20 069 | 100 | Spring to Summer | NW | Shallow to Medium | Sand | [44] |
| GNS | Pouting gillnets | 6 526 | 60 | Summer | NW | Shallow | Mixed | [45] |
| GNS | Atlantic horse mackerel gillnets | 2 698 | – | – | – | – | – | – |
| GNS | Monkfish gillnets | 2 533 | 220 | Spring to Autumn | S | Shallow to Medium | Sand | [13] |
| GNS | Soles gillnets | 1 692 | 60 | Autumn to Winter | NW; SW; S | Shallow | Sand | [13] |
| GNS | John Dories gillnets | 972 | 200 | Summer | NW | Shallow | Sand and Rock | Interviews (1) |
| GTR | Soles trammel nets | 15 269 | 80-100 | Winter | NW | Shallow | Sand and Mixed | [9] |
| GTR | John Dories trammel nets | 13 273 | 200 | Summer | NW; SW | Shallow | Sand and Rock | On-board observations |
| GTR | Monkfish trammel nets | 11 474 | 220 | Spring to Summer | NW; CW; SW | Shallow to Medium | Sand | [9] |
| GTR | European hake trammel nets | 9 740 | – | – | – | – | – | – |
| GTR | Skates trammel nets | 5 586 | 220 | Summer and Winter | NW | Shallow | Sand | [45] |
| GTR | Turbots trammel nets | 3 070 | 200 | Winter to Spring | NW | Shallow | Sand and Mixed | On-board observations |
| GTR | Common octopus trammel nets | 2 808 | – | – | – | – | – | – |
| GTR | Common cuttlefish trammel nets | 2 665 | 80-100 | Autumn to Winter | NW | Shallow | Sand and Mixed | [13] |
| GTR | Pouting trammel nets | 2 503 | – | – | – | – | – | – |
| GTR | European seabass trammel nets | 1 982 | 80 | Winter | NW; CW | Shallow | Sand and Mixed | Interviews (2) |
| GTR | Atlantic horse mackerel trammel nets | 1 888 | – | – | – | – | – | – |
| LLD | Swordfish drifting longline | 380 | 16/0 and 17/0 | Autumn | CW; SW | Deep | Unk | [46] |

*(Continued)*

**Table 2.** (Continued)

| Gear | Name of métier | Number of hauls | Mesh size/hook number | Season | Fishing area | Depth | Sediment | Validation through |
|------|---------------|-----------------|----------------------|--------|--------------|-------|----------|-------------------|
| LLD | Blue shark drifting longlines | 310 | 16/0 and 17/0 | Summer | NW | Shallow and Deep | Sand and Unk | [46] |
| LLD | Black scabbardfish drifting longlines | 149 | – | – | – | – | – | – |
| LLS | Black scabbardfish bottom longlines | 15 652 | 5–8 | All year | CW | Deep | Unk | [47] |
| LLS | European conger bottom longlines | 990 | 3–8 | Summer | NW | Shallow and Deep | Rock | Interviews (6) |
| LLS | Wreckfish bottom longlines | 954 | 6–7 | Spring to Summer | CW | Medium to Deep | Sand and Unk | [48] |
| LLS | Silver scabbardfish bottom longlines | 849 | 9 | Summer | CW | Medium | Sand and Rock | Interviews (2) |
| LLS | Blackspot seabream bottom longlines | 810 | 8–9 | Autumn to Winter | CW | Medium | Sand and Rock | On-board observations |
| LLS | Smooth-hound bottom longlines | 598 | – | – | – | – | – | – |
| LLS | European hake bottom longlines | 590 | 7–9 | Spring | NW; CW | Shallow to Medium | Mud and Sand | [49] |
| LLS | Red porgy bottom longlines | 482 | 8–9 | Summer to Autumn | CW | Shallow | Sand and Rock | Interviews (1) |
| LLS | Blackbelly rosefish bottom longlines | 387 | 3–5 | Spring to Summer | CW; SW; S | Medium to Deep | Sand and Unk | Interviews (4) |
| LLS | Skates bottom longlines | 300 | – | – | – | – | – | – |
| LLS | Alfonsinos bottom longlines | 214 | 6 | Winter to Spring | CW; SW | Medium to Deep | Sand and Unk | Interviews (1) |
| LLS | Forkbeard bottom longlines | 178 | – | – | – | – | – | – |

Métiers were also found to be associated with sediment type. Traps were often used in mixed sediments, whereas drifting or bottom longlines were used in sandy and rocky bottoms. The métiers most strongly associated with certain sediments are pouting (*Trisopterus luscus*) traps and gillnets in mixed sediments, monkfish trammel nets and gillnets in muddy sand, and bottom longlines for silver scabbardfish and red porgy in rocky areas (S2 Table).

Regarding spatial differences, the main finding suggests that some métiers that target the same species use different fishing areas, this is the case of hake, monkfish and to a less extent, John Dories. Gillnet and bottom longline fisheries for hake are prevalent in the northwest and the central west, respectively. Hake gillnets are primarily used between Nazaré and Peniche, with high fishing pressure in the Nazaré Canyon, but expanding to Sagres, in the south, and Viana do Castelo, in the north; the longline fishery for hake operates far from the coast, in deep waters to the west of Peniche. In contrast, bottom gillnets for monkfish are used in the southwest and south areas, between Sesimbra and Vila Real de Sto António, while trammel nets for monkfish and John Dories operate mainly in the center west and southwest, between Peniche and Sagres. Trap and gillnet fisheries for pouting, and bottom longlines for European conger are exclusively found in the northwest (Fig 2).

## 4. Discussion

The Portuguese multi-gear fishing fleet consists of around 500 vessels; in this study, only those with electronic logbooks were used, corresponding to 1/3 of the fleet, and measures over nine meters in length. They mainly use nets, traps, and

 

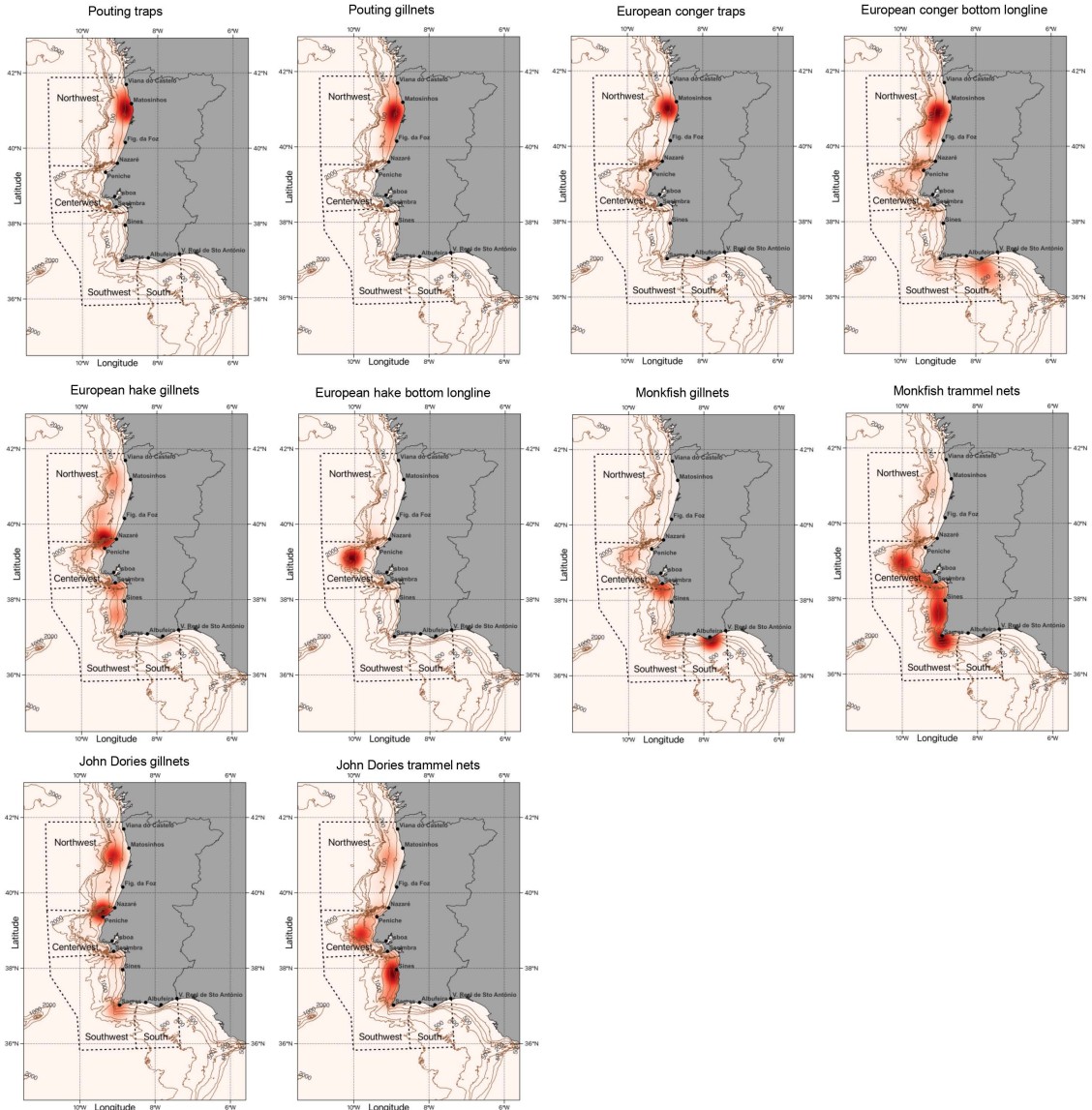

**Fig 2. Heat maps of the fishing pressure of 10 identified métiers.** Selected target species were pouting (*Trisopterus luscus*), European conger (*Conger conger*), European hake (*Merluccius merluccius*), monkfish (Lophiidae), and John Dories (Zeidae). The data combines information from 2014 to 2023.

longlines to catch a wide variety of commercially exploited species. In this study, a total of 28 métiers were identified and validated, many of which for the first time (to the best of our knowledge) at Level 6 [19]. This is particularly important in a multi-gear fleet such as the one studied here, where technical interactions occur, and competitive relationships exist between different fishing fleets and gears exploiting the same stocks. These interactions can be complex, involving direct competition with the same species and the capture of non-target species [16]. This applies to the European hake, pouting, and monkfish, which can be caught using the same gear, but with different selectivity characteristics, resulting in varying levels of by-catch. Separating fishing effort across different gears targeting the same species is crucial for effective fisheries management, since the actions of one fleet impact the catch and profitability of others.

The distribution of hauls for each of the 28 validated métiers showed a strong association with variables such as season, fishing area, depth, and sediment type. Traps and nets were mostly used in the northwest area, in mixed sediments, mud, or sand, whereas longlines mostly operated in the central west area, in steeper slopes on sandy and rocky bottoms. During the interviews, fishers consistently reported that they avoid setting nets and traps in rocky areas due to the high risk of these gears becoming entangled in the seabed. One difference between the trap/net and longline fisheries is that the former has season-specific métiers, such as the common cuttlefish and European seabass trammel nets, restricted to winter. In contrast, bottom longline fisheries are spread more evenly across seasons. We hypothesize that these differences may be related to variations in species availability in distant and close-to-shore waters. Traps and gillnets are primarily used in shallow waters close to the shore. The main target species for these métiers, such as hake, soles, and cuttlefish, are known to migrate between coastal and deeper waters for reproduction [9,50]. Consequently, the availability of fish varies by season depending on the fishing methods used. This is the case for hake, which is targeted with longlines in the spring and with gillnets mainly in the summer.

During the 10 year-period in study, three métiers gained importance: the Norway lobster trap fishery, the trammel net fishery for turbots, and the swordfish drifting longlines fishery. The trammel net fishery, targeting turbot (*Scophthalmus maximus*) and brill (*Scophthalmus rhombus*), increased in line with turbot landings during the same period, with an increase of approximately 300% in weight and 400% in value [15]. The growing interest in this fishery may result from the fact that both turbot species are among the ten with the highest value per kilogram, landed in Portugal [15]. We found that the turbots fishery is mostly associated with seven vessels that alternate their activity between octopus traps and soles trammel nets, indicating that turbots are now the target species resulting in a new métier. The number of hauls also increased in the swordfish drifting longlines fishery; however, this is fishery mainly operates in oceanic waters out of the area of interest of this study [48]. With respect to the Norway lobster trap fishery, it explores a high value niche market, selling live lobsters with increasing demand [51].

Of the nine métiers that were not validated, we propose that seven resulted from misidentification of the target species. The target species was the one associated with the highest-value catches within a given haul. In some cases, a common by-catch species may have achieved the highest commercial value and been mistakenly identified as the target species. This was the case of Atlantic horse mackerel (*Trachurus trachurus*), frequently present in European hake and pouting gillnets [52], and common octopus (*Octopus vulgaris*), a by-catch in soles and cuttlefish trammel nets [50]. Similarly, forkbeards and skates are among the top ten most economically important species/groups on bottom longlines. Given their economic importance, at the initial stage of our procedure, they were identified as potential target species. This underlines the importance of the validation process of métiers.

For the remaining two unvalidated métiers – drifting longlines for black scabbardfish (*Aphanopus carbo*) and bottom longlines for smooth hound (*Mustelus mustelus*) – we assumed that the gear registered in the electronic logbook was incorrect. We compared the entries in the electronic logbooks for drifting longlines and bottom longlines for black scabbardfish. The drifting longline hauls for black scabbardfish come from one vessel in 2019; this vessel used bottom longlines for black scabbardfish in all other years. The species composition, catches, and fishing areas are identical to those of bottom longlines targeting black scabbardfish on all other vessels. Although this was not confirmed with the skipper of the relevant vessel, we hypothesize that it most likely resulted from a systematic error in the gear selected when filling in the electronic logbooks. Similarly, the bottom longline fishery for smooth hound could not be validated. According to our data, the last fishing hauls were in 2020, at depths of around 1500 meters, which is inconsistent with the biology of smooth hound, as it is a demersal shark that inhabits depths of around 50 meters and up to 600 meters [53]. Reports of misidentification of deep-water sharks were found until 2016, where deep-water longliners caught leafscale gulper shark (*Centrophorus squamosus*) and Portuguese dogfish (*Centroscymnus coelolepis*) but reported as school shark (*Galeorhinus galeus*) [54].

Regarding the trammel nets fishery targeting soles and cuttlefish, although it has been studied before [9,13], further research is needed to determine whether it represents a single or distinct métiers. Both species are targeted with trammel

nets of 80–100 mm mesh size, in sandy bottoms, mainly in the northwest area. Researchers studied the trammel net fishery for soles and cuttlefish and showed that while soles are available to the fishery all year round, cuttlefish availability is lower, between spring and summer, possibly due to their inshore-offshore migrations [50]. This is in line with the results presented here. The methodology used here may have led to the identification of one métier where the cuttlefish is less abundant (identifying soles as the target species) and another one when the cuttlefish is more abundant (identifying cuttlefish as the target species).

A total of ten species of skates is landed in mainland Portuguese fishing ports [45]. In this study, all these species were included in a single group, the family Rajidae. This is because it has been demonstrated that most ray species are misidentified during landings [55]. The most important species of skates in terms of weight landed are blonde skate (*Raja brachyura*) at 52%, thornback skate (*Raja clavata*) at 22%, spotted skate (*Raja montagui*), and undulate skate (*Raja undulata*), both at 7% [55]. According to IUCN, three of these species (*R. brachyura, R. clavata, and R. undulata*) are Near Threatened, while *R. montagui* is of Least Concern. Of the remaining six species identified in 2020 [45] two are Least Concern (*Leucoraja naevus* and *Raja miraletus*), one is Near Threatened (*Raja microocellata*), one is Vulnerable (*Dipturus oxyrinchus*), and two are Endangered (*Dipturus oxyrinchus* and *Leucoraja circularis*) [56]. Since 2017, the national sampling program, responsible for collecting, managing, and analyzing biological data essential to the Common Fisheries Policy and for assessing the state of marine ecosystems – developed a statistical procedure to correct the misidentification of skates in landings data [54,57]; nevertheless, efforts should be made to improve the identification by landing personnel in national fishing ports, ultimately to implement conservation measures for the most vulnerable species.

This study was based on approximately 150 vessels, accounting for about one-third of the multi-gear coastal fleet. Information on the remaining fleet is unavailable because, for smaller vessels, the DGRM only requires handwritten logbooks, which were not analyzed here. Previous work compared the catch compositions, for the same main gear, in coastal vessels with and without electronic logbooks and found no significant differences between the two groups [12]. Therefore, we are confident that the fleet analyzed in this study is a representative sample of the entire coastal fleet.

Analyzing electronic logbooks is crucial for characterizing and estimating fishing effort, producing scientific advice in support of the management of the EU's multi-gear fleets. The detailed information about métiers down to level 6, their gear characteristics, and temporal and geographic fishing effort distribution associated with each vessel provides valuable data for improving the efficiency of sampling programs. In fact, with few exceptions, métiers cannot be assigned individual vessels but are rather a reflection of the activity undergone by each vessel at a specific time in a certain area. The common situation is to have groups of vessels switching between métiers, as they target different components of the fish assemblages at different times.

Fishery-based information, as it has been used within the scope of ICES, has not considered the existence of these directed fisheries; their integration as stratification levels in sampling programs is crucial to obtaining more accurate species-specific indices of abundance, fishing effort, and discards, which can be used in stock assessment. Moreover, applying this knowledge to fisheries management improves the balance between the exploitation and conservation of marine resources in coastal areas, especially when complemented by the ecological footprint of each gear (e.g., bycatch and capture of protected species). At the regional level, management can entail closing areas and seasons to specific gears while still allowing fishing activity in other areas and periods. This approach provides information to protect biodiversity, conserve exploited resources, and maintain the economic viability of multi-gear coastal fleets. By 2030, all EU fishing vessels will be required to implement electronic logbooks and monitoring devices [58]. To enhance the quantity and quality of fisheries-dependent data, we recommend including additional details such as gear length, soaking times, and mesh/hook sizes – starting on métiers with higher ecological footprints, such as those using gillnets and trammel nets.

## Supporting information

**S1 Table. List of 307 potential métiers.** Includes: main gear (DRB – bivalve dredges, FPO – traps, GNS – gillnets, GTR – trammel nets, LLD – drifting longline, and LLS – bottom longline), scientific name of the target species, number of vessels, number of trips, and number of hauls for the study period (2014–2023).
(CSV)

**S1 Fig. Histograms by main gear show the frequency of hauls for each potential métier, in descending order.** The main gears represented are: DRB – bivalve dredges, FPO – traps, GNS – gillnets, GTR – trammel nets, LLD – drifting longline, and LLS – bottom longline. Only up to 20 métiers of each gear are represented. The vertical dashed line is the cut-off of the point at which the graph curvature stabilized and was used to decide which métiers would be considered for validation.
(PDF)

**S2 Table. List of the 28 validated métiers.** Includes: code for main gear used (DRB – bivalve dredges, FPO – traps, GNS – gillnets, GTR – trammel nets, LLD – drifting longline and LLS – bottom longline), name of the métier (combining the main gear and the target species), scientific name of the target species, number of hauls and levels of the variables that were used to characterize the métier: year (2014−2023), season (spring – apr to jun; summer – jul to sep; autumn – oct to dez; and winter – jan to march), fishing area (northwest, center west, southwest and south), depth (shallow – less than 200 meters, medium – 200–500 meters and deep – more than 500 meters), and sediment type (mud, muddy sand, sandy mud, sand, coarse grained sediment, rock and boulders, mixed sediment and unknown). The values in the body of the table, for the variables used to characterize the métier, correspond to the percentage of hauls in each level of the variable (years 2014−2023). Within each variable, the colors represent the importance of the level (white: 0%−25%, light green: 25%−50%, intermediate green: 50%−75%, and dark green: 75%−100%).
(XLSX)

**S3 Table. p-values for the chi-square tests for uniform distribution (Zone, Season, and Depth), for the test of independence between Zone and Season (ZonexSeason), and for the linear trend for the number of hauls in a year (Year).** The Bonferroni correction for multiple tests was applied within each variable. The trend in the number of hauls per year was analyzed using linear regression; significant values are highlighted in green, and all slopes were positive. NA corresponds to situations where frequencies were less than five in one or more classes (no chi-square tests were applied).
(XLSX)

**S1 File. R script file.** Script file for the analysis and graphical representations of the results.
(R)

## Acknowledgments

The bathymetric metadata and Digital Terrain Model data products have been derived from the EMODnet Bathymetry portal - http://www.emodnet-bathymetry.eu. The authors thank the Directorate-General for Natural Resources, Safety and Maritime Services, and the fishers interviewed in this work for the data provided.

The authors would like to thank the contributions of three anonymous reviewers who significantly helped to improve this study.

## Author contributions

**Conceptualization:** Pedro Leitão, Margarida Castro, Aida Campos.

**Data curation:** Pedro Leitão.

**Formal analysis:** Pedro Leitão.

**Funding acquisition:** Aida Campos.

**Investigation:** Pedro Leitão.

**Methodology:** Pedro Leitão.

**Project administration:** Aida Campos.

**Resources:** Aida Campos.

**Software:** Pedro Leitão.

**Supervision:** Margarida Castro, Aida Campos.

**Validation:** Pedro Leitão.

**Visualization:** Pedro Leitão.

**Writing – original draft:** Pedro Leitão.

**Writing – review & editing:** Pedro Leitão, Margarida Castro, Aida Campos.

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
