## [Decision Letter · Decision Letter 0]

24 Nov 2025

PONE-D-25-41485Identification of métiers in a multi-gear, multi-species fisheryPLOS ONE

Dear Dr. Leitão,

Thank you for submitting your manuscript to PLOS ONE. After careful consideration, we feel that it has merit but does not fully meet PLOS ONE’s publication criteria as it currently stands. Therefore, we invite you to submit a revised version of the manuscript that addresses the points raised during the review process.

We look forward to receiving your revised manuscript.

Kind regards,

Aldo Corriero, Ph.D.

Academic Editor

PLOS ONE

Journal Requirements:

1.Please ensure that your manuscript meets PLOS ONE’s style requirements, including those for file naming. The PLOS ONE style templates can be found at

“This study received Portuguese national funds from FCT - Foundation for Science and Technology through projects

UIDB/04326/2020 (DOI:10.54499/UIDB/04326/2020),

UIDP/04326/2020 (DOI:10.54499/UIDP/04326/2020) and

LA/P/0101/2020 (DOI:10.54499/LA/P/0101/2020), P. Leitão was the recipient of a PhD scholarship 2022.11214.BDANA.”

“NO”

5. We note that Figures 1 and 2 in your submission contain map/satellite images which may be copyrighted. All PLOS content is published under the Creative Commons Attribution License (CC BY 4.0), which means that the manuscript, images, and Supporting Information files will be freely available online, and any third party is permitted to access, download, copy, distribute, and use these materials in any way, even commercially, with proper attribution. For these reasons, we cannot publish previously copyrighted maps or satellite images created using proprietary data, such as Google software (Google Maps, Street View, and Earth). For more information, see our copyright guidelines: http://journals.plos.org/plosone/s/licenses-and-copyright.

1. You may seek permission from the original copyright holder of Figures 1 and 2 to publish the content specifically under the CC BY 4.0 license.

Additional Editor Comments:

The reviewers noted shortcomings in both the methodological approach and the presentation of the results. Quantitative data should be provided to substantiate general statements in the Results section, and the validation criteria require clearer specification. The statistical analysis also warrants further improvement. Please make sure that all comments raised by the reviewers are comprehensively addressed in the revised manuscript.

Reviewers' comments:

Reviewer’s Responses to Questions

**Comments to the Author**

1. Is the manuscript technically sound, and do the data support the conclusions?

Reviewer #1: Yes

Reviewer #2: Yes

Reviewer #3: Partly

Reviewer #4: Partly

2. Has the statistical analysis been performed appropriately and rigorously? 

Reviewer #1: Yes

Reviewer #2: N/A

Reviewer #3: No

Reviewer #4: No

3. Have the authors made all data underlying the findings in their manuscript fully available?

Reviewer #1: Yes

Reviewer #2: No

Reviewer #3: Yes

Reviewer #4: No

4. Is the manuscript presented in an intelligible fashion and written in standard English?

Reviewer #1: Yes

Reviewer #2: Yes

Reviewer #3: Yes

Reviewer #4: Yes

5. Review Comments to the Author

Reviewer #1: The findings are significant as they clearly demonstrate the complexity of quantifying fishing effort in mixed fisheries and the role of technical interactions and competition between gears. Importantly, the study highlights the growing value of digital fisheries data in improving the robustness of fisheries management. The validation of métiers using multiple sources adds credibility to the results, while the regional, seasonal, and temporal analyses provide additional depth to the understanding of fleet dynamics.

The manuscript is clear, well-structured, and methodologically sound. The results are presented logically and supported by evidence. Overall, the study makes a meaningful contribution to fisheries management and policy, particularly in the European context but with relevance to other multi-gear fisheries worldwide.

Reviewer #2: This manuscript presents defines a procedure to allocate fishing hauls to metiérs, based on the Portuguese multi-gear coastal fleet. In turn, the metiérs are validated through a combination of literature studies and interviews. The manuscript is generally very well presented. I have the following minor comments:

1) The authors define the target species based on the landed value. However, the authors note that for some of the non-validated metiérs this did not work, because a landed bycatch was much more valuable. Therefore, I wondered how different the validated metiérs would be if the target was defined based on landed weight, and would that have allowed more metiérs to be validated?

2) In the validation of metiérs, it is currently unclear to me how many skippers were asked per metiér? And, if more than one was asked, what would happen if some says "yes" and some say "no"? Further, it was not clear to me how many of the non-validated metiérs that were due to "no" answers and how many that could not get an answer?

3) My understanding is that the new procedure proposed defines a finer scale /lower level metiér than what has previously been used. I think it would be illustrative to compare the new procedure to what was previously done - for example by adding the previous metiér in table 2.

Reviewer #3: In general, this manuscript is informative and an impressive effort to summarize activities of a large and diverse fishing fleet over the past decade along the coast of Portugal. However, I think the manuscript requires important revisions before being considered for publication. My primary concerns are that results and discussion are very descriptive and lacking specificity and relevance. I also have some concerns about the statistical analyses and recommend that the authors evaluate a more robust statistical approach. Below are some specific comments and suggestions.

L13-29 Abstract: Include results in the abstract. The abstract only includes methods and a general conclusion that is not unique to your study. It is important for your abstract to include a summary of your primary results and conclusions that are unique to your study, including dominant fisheries and the seasonal and regional trends noted as objectives. Also, consider using first person more frequently throughout, including in the abstract. For example, “We used electronic log books….”, rather than “This study uses…”

L101: Métier should be defined earlier when this term is first used.

L183: Did you specify which months were included in each season and accurately define all variables?

L233-235, L287-293, and throughout: Please describe the relevant results found in these tables rather than simply telling us what information is in the tables - that is the purpose of the table caption and does not need to be repeated in the text.

L257: Chi-Square tests are sensitive to sample size and multiple tests even with Bonferroni corrections. Large sample sizes often produce highly significant results despite having small effect sizes that are potentially irrelevant. For example, Table S3 has nearly all p-values <0.001 even though some within category differences are much smaller than others. A specific example of this is season for common octopus traps (relatively small differences among seasons) and European sea bass trammel nets (large difference among seasons). Yet, both of these have p < 0.001, which is likely due to the extremely large sample size of hauls for octopus traps, but which also questions the sample size bias in the results. Your current statistical approach treats every haul as independent, which is not true because you have pseudo-replication from repeated measures of the same vessel, year, etc. when assessing differences among season. Did you consider a more robust and flexible modeling approach such as Poisson regression?

L280: What was the percent accuracy of the métiers that were validated? This is important to inform how accurate the remaining nine were. You do speak to the potential cause of non-validation on L423-424, which is confusing since this topic was not mentioned until then.

L298: What is meant by most representative? Most number of vessels, most valuable, etc.?

L300: Here and elsewhere, species or taxa are mentioned without including scientific names or genera/families where appropriate. When you first mention species groups in the body of the manuscript, you should direct the reader to the supplementary tables where the scientific names are included.

L316 - 336: Throughout these paragraphs, quantitative values should be included within text to support statements of “almost exclusively”, “often”, “strongly associated”, etc. and then parenthetically reference a table where the reader can find more quantitative results. Also, include the depth you are referring to when mentioning “shallow”

L338: Similar to my comment above, please do not start a paragraph or sentence simply stating what information is in a figure or table.

L477: Likely replace “fixed the” with “reduced”. It is unlikely that all misidentifications were “fixed” so they no longer occur, correct?

L515 – 518: If this is a requirement already in place by the EU, then this concluding statement is not informative, but is instead simply restating a known fact published elsewhere. It would be more far more informative to use results from your study to provide recommendations of which fisheries are most lacking of information and should be prioritized for this effort – or any other recommendations learned from your study that are not already described elsewhere.

L727: There is no description of what the vertical line in each panel refers to

Reviewer #4: PONE-D-25-41485

Identification of métiers in a multi-gear, multi-species fishery

Pedro Leitão, Margarida Castro, Aida Campos

The manuscript presents an approach to identify metiers within the Portuguese multi-gear fishery and examines the variation of fishing effort by métier over time, region and other factors such as depth and bottom type. This is a crucial step to improve management for a fishery which has high socio-economic importance and is highly diverse in terms of fleet, target species, seasonal and spatial operation, etc and therefore cannot be managed as an homogenous fishery.

With respect to criteria valued by PONE:

1. The study presents the results of original research.

Yes (to my knowledge)

2. Results reported have not been published elsewhere

No (to my knowledge)

3. Experiments, statistics, and other analyses are performed to a high technical standard and are described in sufficient detail.

No. The validation of métiers is not clear. The authors mention they used “previous studies, onboard data observations, and interviews with fishers” although they do not explain which type of validation and how was the information used for each métier. Were the answers to the interviews with skippers pooled ? How? What onboard data observations were used and how was the information combined with interviews?

Moreover, the approach for the identification of métiers appears to be based on single target species which seems to contradict the multispecies targeting nature of this kind of fisheries and neglect results from previous studies by the same authors. In fact, from the ca. 300 métiers initially identified, the authors end validating 28 métiers suggesting that the initial groups were not true métiers. Although the approach to select métiers is acceptable it is also subjective. I suggest to evaluate the sensitivity of the results to the cutting point to ensure that the results are robust.

The variation of fishing effort with year, season, fishing area, depth, and sediment is not fully supported by a statistical analysis. The authors apply homogeneity tests using variables separately neglecting potential interactions between them (which are likely, for example, between fishing area, depth and sediment type). Moreover, the comparisons between levels of the variables are not statistically tested (statements such as “effort increased over time” are not supported. Authors could explore GLMs to perform the analyses and overcome this gap.

4. Conclusions are presented in an appropriate fashion and are supported by the data.

Depending on the revision of the methods, conclusions may need to be revised.

5. The article is presented in an intelligible fashion and is written in standard English.

Generally yes although the English needs to be improved in some sections.

6. The research meets all applicable standards for the ethics of experimentation and research integrity.

Yes.

7. The article adheres to appropriate reporting guidelines and community standards for data availability.

Yes

Please find minor comments directly on the manuscript attached.

6. PLOS authors have the option to publish the peer review history of their article (what does this mean?). If published, this will include your full peer review and any attached files.

Reviewer #1: No

Reviewer #2: No

Reviewer #3: No

Reviewer #4: No

---

## [Author Response · Author response to Decision Letter 1]

7 Apr 2026

Dear reviewers,

The authors would like to thank the contributions which significantly helped to improve this study. Please find the comments for each of the raised questions in the reviewers letter.

Best regards,

---

## [Decision Letter · Decision Letter 1]

15 Apr 2026

Identification of métiers in a multi-gear, multi-species fishery

PONE-D-25-41485R1

Dear Dr. Leitão,

We’re pleased to inform you that your manuscript has been judged scientifically suitable for publication and will be formally accepted for publication once it meets all outstanding technical requirements.

Kind regards,

Aldo Corriero, Ph.D.

Academic Editor

PLOS One

Additional Editor Comments (optional):

All the reviewers' comments have been properly addressed and the manuscript can be accepted for publication.

Reviewers' comments:

Reviewer’s Responses to Questions

**Comments to the Author**

1. If the authors have adequately addressed your comments raised in a previous round of review and you feel that this manuscript is now acceptable for publication, you may indicate that here to bypass the “Comments to the Author” section, enter your conflict of interest statement in the “Confidential to Editor” section, and submit your "Accept" recommendation.

Reviewer #2: All comments have been addressed

2. Is the manuscript technically sound, and do the data support the conclusions?

Reviewer #2: Yes

3. Has the statistical analysis been performed appropriately and rigorously? 

Reviewer #2: Yes

4. Have the authors made all data underlying the findings in their manuscript fully available?

Reviewer #2: No

5. Is the manuscript presented in an intelligible fashion and written in standard English?

Reviewer #2: Yes

6. Review Comments to the Author

Reviewer #2: (No Response)

7. PLOS authors have the option to publish the peer review history of their article (what does this mean?). If published, this will include your full peer review and any attached files.

Reviewer #2: No

---

## [Editor Report · Acceptance letter]

PONE-D-25-41485R1

PLOS One

Dear Dr. Leitão,

I'm pleased to inform you that your manuscript has been deemed suitable for publication in PLOS One. Congratulations! Your manuscript is now being handed over to our production team.

Kind regards,

on behalf of

Dr. Aldo Corriero

Academic Editor

PLOS One